# Fast and Accurate Prediction of Refractive Index of Organic Liquids with Graph Machines

**DOI:** 10.3390/molecules28196805

**Published:** 2023-09-26

**Authors:** François Duprat, Jean-Luc Ploix, Jean-Marie Aubry, Théophile Gaudin

**Affiliations:** 1Molecular, Macromolecular Chemistry and Materials, ESPCI Paris, PSL Research University, 75005 Paris, France; jean-luc.ploix@espci.psl.eu; 2Unité de Catalyse et Chimie du Solide, Centrale Lille, University Lille, UMR CNRS 8181, 59000 Lille, France; jean-marie.aubry@univ-lille.fr; 3Dassault Systemes BIOVIA, Cambridge CB4 0FJ, UK; theophile.gaudin@3ds.com

**Keywords:** refractive indices, graph machines (GM), machine learning, structured data, stereochemistry, Docker, COSMO-RS

## Abstract

The refractive index (RI) of liquids is a key physical property of molecular compounds and materials. In addition to its ubiquitous role in physics, it is also exploited to impart specific optical properties (transparency, opacity, and gloss) to materials and various end-use products. Since few methods exist to accurately estimate this property, we have designed a graph machine model (GMM) capable of predicting the RI of liquid organic compounds containing up to 16 different types of atoms and effective in discriminating between stereoisomers. Using 8267 carefully checked RI values from the literature and the corresponding 2D organic structures, the GMM provides a training root mean square relative error of less than 0.5%, i.e., an RMSE of 0.004 for the estimation of the refractive index of the 8267 compounds. The GMM predictive ability is also compared to that obtained by several fragment-based approaches. Finally, a Docker-based tool is proposed to predict the RI of organic compounds solely from their SMILES code. The GMM developed is easy to apply, as shown by the video tutorials provided on YouTube.

## 1. Introduction

The refractive index (n) of a given medium is the ratio of the speed of light in a vacuum to the speed of light in that medium. It is one of the most important optical parameters of molecular compounds and materials and is widely exploited in physics, biology, and chemistry. For example, n was routinely measured to characterize organic molecular liquids and confirm their authenticity and purity in much the same way that melting points were determined to characterize molecular solids. Nowadays, these measurements are superseded by more accurate and informative spectroscopic and chromatographic methods.

On the other hand, the optical applications of refractive indices have retained a major interest in the development of innovative materials and formulations. When a light beam passes from one isotropic medium 1 to another 2, a part of the light is reflected and the other part is refracted. Knowledge of their refractive indices *n*_1_ and *n*_2_ allows one to calculate the relationship between the angles of incidence *θ*_1_ and refraction *θ*_2_ according to the so-called Snell–Descartes’s law (Equation (1)).
*n*_1_ sin *θ*_1_ = *n*_2_ sin *θ*_2_(1)

Thus, if a light beam is sent at the interface 1/2 with a given angle of incidence *θ*_1_, knowledge of *n*_1_ and *n*_2_ allows predicting *θ*_2_. In particular, when the medium 1 is air, whose refractive index is very close to 1, the accurate measurement of *θ*_2_ provides the refractive index of a liquid *n*_2_ by applying Equation (1). Given the ease of such measurements, the refractive indices of thousands of liquids are known with high accuracy (≈10^−3^ with standard refractometers) and are readily available in the literature.

The refractive index has many industrial applications, among which two typical didactic examples are given below. In formulation chemistry, when a formulator wishes to impart particular visual effects to dispersed systems, he strives to minimize or, on the contrary, maximize refractive index differences between dispersed particles and the surrounding matrix. The first case corresponds to the index-matching method, whose principle consists of making the refractive indices *n*_1_ and *n*_2_ coincide. Refraction and reflection phenomena are eliminated, and the light passes through the heterogeneous material as if it were isotropic. For example, to make a transparent toothpaste, the refractive index of the abrasive particles (SiO_2_, x H_2_O) must be equal to the refractive index of the water-based toothpaste matrix. As the refractive index of hydrated silica (*n* ≈ 1.44) is significantly higher than that of water (*n* = 1.333), it is necessary to add to the aqueous phase a very precise amount of an edible liquid with a higher refractive index, such as sorbitol (*n* = 1.525), in order to match the refractive indices of the particles and the matrix [1].

On the contrary, to maximize refraction, the difference between n_1_ and n_2_ must be maximized because, for “natural light” (i.e., unpolarized light), the intensity of specular reflection increases with the difference (*n*_1_ − *n*_2_) according to Fresnel’s law of reflection and Schilck’s approximate equation (Equations (2) and (3)):(2)$$R(\theta_{1})=R_{0}+(1-R_{0})(1-cos\theta_{1}{)}^{5}$$
(3)$$with\;R_{0}={\left[\frac{n_{1}-n_{2}}{n_{1}+n_{2}}\right]}^{2},$$
where *R*_0_ is the reflection coefficient for light incoming perpendicular to the interface between the two media 1 and 2 and *θ*_1_ is the angle of incidence.

For example, to obtain a white coating with high opacifying power, the refractive index of the dispersed particles and of the polymer matrix forming the film must be as different as possible. This can be achieved by implementing two different strategies. The first consists of introducing into the paint formula a white pigment such as TiO_2_ (rutile form), whose refractive index (*n* = 2.75) is much higher than that of the organic matrix (*n* ≈ 1.48). The second method consists in formulating the liquid paint beyond the CPVC (critical pigment volume concentration) so that, after drying, some microbubbles of air remain inside the film, whose refractive index (*n* = 1.00) is significantly lower than that of the surrounding matrix. The latter strategy is cheaper than the former, but the resulting coating is porous and has low mechanical strength. It is nevertheless suitable for whitening ceilings where the coatings are not subject to mechanical stress [2].

Besides its ubiquitous role in the optical properties of materials and molecules, the refractive index *n* also provides valuable information on the mean polarizability *α* of molecular compounds through the Lorentz-Lorenz relationship (Equation (4)) [3]:(4)$$\frac{n^{2}-1}{n^{2}+2}=\frac{\alpha }{3\varepsilon_{0}V_{m}}\;,\;$$
where the polarizability *α* (C·m^2^·V^−1^) expresses the tendency of the molecule to acquire an electric dipole moment when subjected to an electric field, *ε*_0_ (F·m^−1^ or C·m^−1^·V^−1^) is the vacuum permittivity, and $V_{m}$ (m^3^) is the molar volume. However, in the literature, the Lorentz-Lorenz relationship is more frequently expressed using the polarizability volume *α*’ = *α*/(4π*ε*_0_) instead of the polarizability *α*, which leads to Equation (5):(5)$$\frac{n^{2}-1}{n^{2}+2}=\frac{4\pi \alpha \prime }{3V_{m}},$$
where *α*′ and *V_m_* are expressed in the same unit (m^3^) and have the same order of magnitude.

London showed that the induced dipole-induced dipole interactions that act between all atoms and molecules, including totally neutral ones such as noble gases, are closely linked to their polarizability *α*. Actually, the interaction energy *w*(*r*) (in J) between two identical spherical non-polar molecules is expressed as a function of their polarizability *α* according to Equation (6) [3]:(6)$$w\left(r\right)=-\frac{3}{4}\frac{\alpha^{2}I}{{\left(4\pi \varepsilon_{0}\right)}^{2}r^{6}},$$
where *I* is the first ionization energy of the molecule (in J) and *r* is the intermolecular distance (in m).

London called this type of interaction “dispersion forces” to emphasize that they can be expressed as a function of polarizability, which also appears in light dispersion theory. Dispersive interactions play an essential role in industrially relevant physicochemical phenomena. For example, they are a key component within the Hansen solubility parameters theory, a popular approach commonly used to find solvents and solvent mixtures capable of dissolving a given molecular or macromolecular compound [4,5].

On the other hand, refractive index can be accurately measured experimentally, to the point that it is even used to help identify compounds (typically polymers) using the so-called “refractive index increment” that a dissolved compound generates on a solvent of known refractive index [6]. As a consequence, refractive index, through its direct connection with dispersion energy, could serve as a reliable yardstick for parameterizing those specific forces within any predictive method.

The first approaches that were used to predict the refractive index without the need for experimental data such as molar refraction or molecular volume were group contribution methods [7,8]. Using 38 group increments, Hoshino et al. [8] predicted the refractive index of 377 hydrocarbons and 224 non-hydrocarbons containing heteroatoms such as oxygen, nitrogen, sulfur, and halogens. According to Hoshino, the refractive index of a compound at 20 °C is simply estimated by dividing its molecular weight by a sum of group increments chosen according to the knowledge of its topological formula. With this simple and practical method, they reported an average error of 0.006 and a maximum error of 0.042 for the refractive index prediction of their dataset of 601 compounds. However, as in their first paper [7], the authors did not indicate the dataset used to parameterize their group contributions for the refractive index at 20 °C. One of the drawbacks of this method, pointed out by Hoshino et al., is that it does not distinguish between two isomers that decompose into the same number of groups, which then have the same estimated index. Secondly, although group increments are parameterized for the mentioned heteroatoms, the results are mostly accurate for hydrocarbons. For example, using the provided equation, the estimation of the refractive index of 1,4-dichloro-2-iodobenzene yields an error of 0.067, well beyond the maximum advertised error. In the following decade, Gakh et al. [9] proposed a new computational scheme using graph theory for predicting the refractive index of alkanes. The topological information of 109 alkanes was encoded into Wiener-type structural graph invariants [10] that served as inputs to a feed-forward neural network. Once the network had been trained with this set, the refractive indices of 25 fresh alkanes were predicted with a very good root mean square error (RMSE) equal to 0.003. While limited to a small hydrocarbon set, this new method emphasized the importance of topology in predicting the refractive index as well as the effectiveness of neural networks for such a task. A few years later, Katritzky et al. [11] published the first quantitative structure-property relationship (QSPR) model capable of estimating the refractive index of organic liquids from the information encoded in their 3D chemical structure. This linear model based on quantum chemical, topological, and constitutional descriptors was trained on a set of 125 organic compounds of different classes, providing a root mean square training error (RMSTE) equal to 0.016. It was then applied to an external test set of 25 diverse organic liquids and resulted in a prediction with a computed test RMSE equal to 0.022 and a maximum error of 0.039. The results were not as good as those obtained by Hoshino and Gakh, probably because only five descriptors were used and also because the training set was too small given the variety of chemical functions considered. Still, just like the two previously described approaches, this method has the advantage of not relying on experimental parameters; predictions are made solely on the basis of molecular structure. Soon after, Cocchi et al. [12] described similar QSPR models for predicting five different physicochemical properties. For refractive index prediction, a model with 20 descriptors was trained on a dataset of 67 organic solvents, resulting in a computed RMSTE equal to 0.013. Applied to the prediction of the refractive index of 29 solvents, this model led to a test RMSE equal to 0.018, with a maximum observed error of about 0.050. Despite the much larger number of descriptors than for Katritzky’s QSPR model, the improvement in fit was rather small, especially since the Katristky 125-liquid dataset contained more complex molecules, e.g., with multiple cycles. Several other QSPR models based on molecular descriptors, limited to sets of a few hundred compounds, have been proposed for refractive index prediction [13,14,15,16,17]. For datasets that are not only hydrocarbons, the best results were obtained with a model based on associative neural networks. With a network of eight input descriptors and seven hidden neurons, the refractive index of a test set of 28 compounds was predicted with an RMSE equal to 0.010 [16].

Gharagheizi et al. were the first authors to develop a model with a large dataset for the estimation of the refractive index of pure compounds [18]. Thanks to a collection of 80 chemical substructures determined using a database of 9536 mostly organic compounds, the refractive index of a new molecule was estimated by summing the contributions of the substructures composing it. RMSE values equal to 0.020 were obtained for the validation and test datasets, both consisting of about 1000 molecules and used, respectively, to assess the validity and predictive capability of the model. Yet, nearly 200 molecules (2%) were listed as outliers, being estimated by the model with a deviation from the measured value greater than 0.060. Despite this, the model can undoubtedly be improved because, among the structures qualified as outliers, 36% were not in the liquid state at 20 °C, and 30% of the remaining liquids had a reported experimental refractive index differing by more than 0.050 from verified values.

According to Cai et al. [19], one possible explanation for the inaccuracies of the refractive index predictions reported in the above-mentioned papers is the lack of a physical basis for the approaches used. In an attempt to develop a more accurate model, these authors approximated the Lorentz-Lorenz Equation (5) by expressing the ratio *α′*/*V_m_* of a compound as a sum of the ratios $\alpha_{i}^{\prime }$/*V_m,i_* of all its functional groups *i* (Equation (7)):(7)$$\frac{n^{2}-1}{n^{2}+2}=\frac{4\pi }{3}\frac{\alpha \prime }{V_{m}}\approx \frac{4\pi }{3}\sum_{i}x_{i}\frac{\alpha_{i}^{\prime }}{V_{m,i}},$$
where $x_{i},\;\alpha_{i}^{\prime }\;and\;V_{m,i}$ are the mole fraction, the polarizability volume, and the molar volume of each group *i* respectively. To process efficiently all the available data, Cai et al. had to split their database into three training sets, for which they introduced three separate group contribution models with 32, 27, and 25 parameters, respectively. When estimating the refractive index of the 234 compounds in their training sets, a training RMSE equal to 0.016 was obtained. Yet, application of their models to a fresh set of 106 simple molecules from a paper published by Redmond et al. [17] led to a much larger test RMSE (0.046).

For Bouteloup and Mathieu, who estimated the refractive index of a large set of 7243 compounds [20], these results were due to the fact that the ratio α′/$V_{m}$ in Equation (5) does not obey additivity principles, whereas it does for each of the α′ and $V_{m}$ terms. Thus, with the idea of using similar additivity procedures for both terms, the authors proposed Equation (8), in which the molar polarizability volume α′ from the first equality of Equation (7) is equivalently replaced by the molar refractivity $R_{D}$ (equal to 4π*α*′/3):(8)$$\frac{R_{D}}{V_{m}}=\frac{n^{2}-1}{n^{2}+2}=\frac{\sum_{i}x_{i}R_{i}}{\sum_{i}x_{i}V_{i}}\;,$$
where $x_{i}$, $V_{i}$ and $R_{i}$ are the mole fraction and contributions assigned to each nonhydrogen atom *i* of a compound of refractive index *n* [21]. In this so-called geometrical fragment (GF) approach [22], a molecule is split into atomic fragments that contribute to the molar refractivity $R_{D}$ and molar volume $V_{m}$ according to three parameters: the atomic number $Z_{i}$ of the atom, its coordination number $n_{i}$ and the number $n_{H_{i}}$ of hydrogen atoms among its $n_{i}$ neighbors. In order to parameterize the GF method for the chosen refractive index training set consisting of 3622 compounds (out of 7243), the authors proceeded as follows: (i) they computed the molar volume for the 3622 compounds using 43 molar volume contributions $V_{i}$ previously evaluated by multilinear least-squares regression (MLR) on molar volume [23], (ii) derived the corresponding refractivities $R_{D}$, according to the first equality of Equation (8), and (iii) fitted a set of 46 refractivity increments $R_{i}$ from an MLR against the molar refractivities computed in step ii. The prediction of the refractive index for a new compound can then be computed using the second equality of Equation (8), provided that all the atomic contributions $R_{i}$ and $V_{i}$ needed to describe this new compound are part of the 89 parameters on which the GF model depends. The main drawback of this model stems from this last remark. For example, it is not possible to estimate the refractive index of a compound such as phenylphosphine, as the $R_{P32}$ parameter corresponding to the refractivity increment for a phosphorus bonded to three atoms, two of which are hydrogens, is not defined. Similarly, it is not possible to compute the index for methyldichloroarsine due to the fact that the model has neither of the two parameters $V_{As}$ and $R_{As30}\;$required for the arsenic atom bonded to three non-hydrogen atoms. Another drawback of this approach is its inability to differentiate between two positional isomers that are predicted with the same index, even though their experimental values are different. For example, *ortho*- and *meta*methylthiophenol have experimental index values of 1.613 and 1.629, respectively, whereas the predicted value is 1.644 for both. The same applies to diastereoisomeric compounds whose indices are estimated with identical values, whereas their measured values are not. Finally, as the GF estimation relies on both molar volumes and molar refractivities estimated from experimental values of density and refractive index, respectively, in the end, two errors accumulate when computing the refractive index of a new compound, and this probably limits the accuracy that can be achieved with this method. Despite these intrinsic limitations, the GF method is quite effective for predicting the refractive index of very different compounds using a fragment-based approach, and to date, it has also produced the best results.

Equation (8) can also be used with even simpler atom-based contributions for polarizability (for example, one contribution for the C atom, one for the H atom, and so on). One model provided with the COSMOtherm software, release 2023 [24,25,26,27] (but unrelated to COSMO-RS theory) is based on this idea. Note that the molar volume is predicted using a ready-made QSPR model that is already part of COSMOtherm rather than fitting on-purpose atomic contributions or using experimental density data. A test carried out in the context of this work and added to the Appendix A gives an idea of the expected accuracy of prediction using such a simple atomic contribution framework.

The importance of topology in the good results described above led us to apply the graph machine approach we have developed [28] to the prediction of refractive index. Also based on the 2D structure of molecules, graph machines have the advantage not only of taking the nature of the atoms into account, like a fragment-based approach, but also of preserving their sequence in the molecular structures, thus allowing the estimation of different values for isomers and even diastereomers. We successfully used this tool to predict physico-chemical properties such as surface tension, viscosity, and equivalent alkane carbon number (oil hydrophobicity), showing in particular that it gives complementary results to those obtained with a COSMO-RS approach based on five σ-moment descriptors [29,30,31]. This study thus has two objectives: to test the ability of graph machines to predict the refractive indices of several thousand organic liquids more efficiently than existing models and, from a more methodological standpoint, to verify the ability of graph machines to handle diastereomers.

Therefore, in the present article, graph machine regression (described in Section 3.3) is used to estimate the refractive index of pure liquids at 20 °C. Models are designed and trained from a database of refractive index, at 20 °C, of 3516 molecules belonging to a wide variety of chemical families and containing carbon, hydrogen, oxygen, nitrogen, halogens, sulfur, phosphorus, silicon, boron, germanium, titanium, tin, and selenium atoms. The generalization ability of the resulting model is assessed by the estimation of the refractive index at 20 °C of 3515 compounds that are not present in the training set. The model’s performance is then compared with that obtained using previous QSPR and group contribution approaches on several test bases. Finally, a graph machine model is trained on a set of 8267 compounds whose refractive indices are measured between 20 and 30 °C. Once validated, it is integrated into a demonstration software version 1.0 package written in Python, which is available for download.

## 2. Results

### 2.1. Graph Machine Model Selection

The selection of the model with the appropriate complexity, given the available data, was done by training the graph machine-based models on the 3516-dataset, as defined in Section 3.4, with an increasing number of MLP hidden neurons. In addition to the computation of the VLOO score, as defined in Equation (12), the Root Mean Square Training Error (RMSTE), which is an indicator of the ability of the model to account for the training data, is also computed according to Equation (9):(9)$$RMSTE=\sqrt{\frac{1}{N_{T}}\sum_{i=1}^{N_{T}}{\left({RI}_{exp.}^{i}-{RI}_{est.}^{i}\right)}^{2}}\;,$$
where $N_{T}$ is equal to 3516, ${RI}_{exp.}^{i}$ is the RI value determined experimentally for molecule *i*, and ${RI}_{est.}^{i}$ is the RI value estimated by the model for molecule *i* at the end of the training. The RMSTE and VLOO score computations are repeated three times for each complexity of the models, so the averages are displayed in Table 1.

The observed continuous decrease of the root mean square training error when the complexity increases (top row) validates the ability of the models to handle the training data. Usually, a minimum value for the VLOO score is obtained for a given complexity. In the present case, no minimum is reached, but the VLOO score hardly decreases beyond a complexity of 24 neurons (second row, 0.006), a behavior that is also confirmed by the small variation of the minimum and maximum deviations from experimental values beyond this complexity (two bottom rows, last two columns). This also indicates that the phenomenon of overtraining is not observed. So, since very similar scores are obtained for 24 and 26 hidden neurons, the most parsimonious of these models, i.e., the model with the lower complexity (24 hidden neurons), noted thereafter as GM24, is selected for testing.

### 2.2. Performance of the Selected Graph Machine-Based Model on TCI Datasets

To assess the performance of the GM24 selected model, the RI predictions for the 3515 molecules in the test set are computed for three different parameter initializations, using for each sequence the ten models (out of 100) with 24 hidden neurons that have the smallest VLOO scores. The means of the resulting three computations are the final predictions for the test set. The overall results are summarized in Table 2.

The computed root mean square errors, respectively equal to 0.003 and 0.006 on the training and test sets (second column, rows 1–2), indicate that the GM24 model performs very well on both sets. As expected, the performances are a bit lower in prediction; however, the RMSE value of 0.006 computed for the test set is the same as the one computed for the training set VLOO score (Table 1, penultimate column, second row). This demonstrates that (i) the VLOO score on the training set provides an accurate assessment of the generalization ability of the model; (ii) increasing the complexity of the model, given the available data, is not necessary; and (iii) the quality of prediction is very good.

The quality of the fit for the training set is also reflected by the minimum and maximum deviations observed for the GM24 model (Table 2, first row, columns 4–5), which are moderate. In fact, only two molecules, 1,1,1-trifluoropentane-2,4-dione and 2-acetylcyclohexan-1-one, shown on the left in Figure 1, have an estimated RI with an error greater than 0.020. Since the measured RIs from the TCI catalog were found to be correct for these molecules after verification [32], deviations from the experimental values could be due to an error in the molecular structures. Actually, it was found in the literature that 1,1,1-trifluoropentane-2,4-dione exists mainly in its *syn*-enol form [33], in which two stabilizing hydrogen bonds can exist, as indicated by the dashed bonds for the two conformations displayed on the top right in Figure 1. A COSMO-RS calculation of the tautomer weights in pure 1,1,1-trifluoropentane-2,4-dione, including the diketone form and the two forms with intramolecular H-bonds displayed in Figure 1, supported that the tautomer with O-H⋯O intramolecular H-bonds is the predominant form (cf. Appendix A, Section D, and Appendix A for details). If the enolic form SMILES with stereochemical labels (CC(/C=C(O[H])/C(F)(F)F)=O) is used for the GM24 model RI computation, an estimated value equal to 1.394 is obtained. This value is much closer to the experimental value of 1.388 than the estimated RI for the dione (*RI_est_*. equal to 1.363). The same computation made without a stereochemical label for the enol form (i.e., SMILES equal to CC(C=C(O[H])C(F)(F)F)=O) leads to a less relevant estimate (*RI_est_*. = 1.405). Similarly, the RI computation for the most stable enol form of 2-acetylcyclohexan-1-one, shown at the bottom right of Figure 1, leads to a value of 1.505, more in line with the measured value (*RI_exp_*. = 1.510). These predictions, made with enol forms, lead to values very close to those measured, indicating that in this case, it would be preferable to input the SMILES of the enol forms for the GM construction of these molecules. This also highlights a very important property of graph machines: the ability to detect anomalies in the data, either in the measured values or in the input codes.

With the exception of the two diketones shown in Figure 1, all other molecules in the training set are estimated with deviations from experimental values of less than 0.020 (i.e., 1.5%).

For the test set, the observed deviations in prediction are a bit larger, as highlighted by the higher MIN and MAX values in Table 2 (−0.051 and 0.036). This increase is often explained by the presence of molecules in the test set that are not represented in the training set, i.e., that have structural features that are not known to the model. Thus, among the dozen compounds for which the prediction error is greater than 0.030, two such compounds, the pentafluoro derivatives of the triazatriphosphinines (a) and (b) represented in Figure 2, result in the largest negative errors, equal to −0.051 and −0.048, respectively. No compound of similar structure, a cycle with three nitrogen and three phosphorus atoms, is present in the training set, which explains the poor prediction in these cases.

The maximum positive deviation (0.036) is obtained for pyridazine (c) shown in Figure 2, and this despite the presence of 3-methylpyridazine in the training set. Moreover, 3-methoxypyridazine and 4-methylpyridazine, also two members of the test set that are structurally related to pyridazine, lead in contrast to RI predictions in line with their experimental values. This unexpected discrepancy for pyridazine is explained this time by the analysis of pyridazine and 3-methylpyridazine graph machines: for pyridazine, the function that outputs the RI estimation has a nitrogen atom type label, while for 3-methylpyridazine it has a carbon atom type label. These small differences in GM structures are sometimes sufficient to explain the observed deviations for similar molecules. In the present case, if the SMILES “[C:1]1=NN=CC=C1” is used as input for the construction of the pyridazine GM, the refractive index estimated by the model becomes equal to 1.516, a value that is much closer to *RI_exp_*., equal to 1.524 (Figure 2c). This trick forces the algorithm to position the root node of the pyridazine graph on a carbon atom rather than on a nitrogen atom, as for 3-methylpyridazine. For the three examples detailed above, an improvement in the prediction would be obtained by increasing the number of structurally related compounds in the training set. Another approach is described in Section 4, in which all the compounds of the test set are incorporated into an extended training database.

As an illustration, RI estimates for molecules in the TCI training set and RI predictions for molecules in the TCI test set are plotted against their measured values in Figure 3. The RMSE computed for the training and test sets is equal to 0.003 and 0.006, respectively, as displayed in Table 2, and the determination coefficients R^2^ are above 0.99 for both sets. Data points for the five molecules discussed above are also shown, as is that for thieno[2,3-*b*]thiophene, which shares with pyridazine the largest positive deviation in prediction.

### 2.3. Head-to-Head Comparison of the Graph Machine-Based Model with Three Models Reported by Other Authors

For comparison, our selected GM24 model is applied to predict the refractive index of liquids in the datasets used by the three models already mentioned and noted thereafter: HR-JT [17], CCAI [19], and CRC [20]. The three models used methods based respectively on QSPR [17], group contributions (GC) [19], and geometrical fragment (GF) [20] approaches to derive an equation allowing the estimation of the refractive index.

#### 2.3.1. Rectification of the Datasets

To ensure the quality of the prediction made by the GM24 model, important preparatory work was done for the three datasets. It consisted in discarding compounds (i) that are also present in the TCI training dataset used to design the GM24 model since a true prediction is expected; (ii) that are duplicates or enantiomers; (iii) that melt above 30 °C or are gaseous below 20 °C; (iv) whose RI were not measured between 20 and 30 °C or at a wavelength equal to 589 nm, unless another experimental value from a reliable source measured at 20 °C can replace it. As a result of these recommendations, the HR-JT and CCAI datasets were shortened from 105 to 52 and from 191 to 116 compounds, respectively. Overall, a few RI values were corrected for the HR-JT and CCAI sets when they differed by more than 0.002 from those obtained from reliable sources. Due to its larger size (1625 compounds), a different approach was applied to verify the CRC dataset. This step is needed since previous authors have indicated that it contains many erroneous RI values [20]. Thus, in addition to checking the melting and boiling temperatures of the compounds in the CRC set, a preliminary estimation of the RI of these compounds was performed with the GM24 model to detect possible errors. When the difference between the estimated and measured values was greater than 0.020 in absolute value, the experimental CRC RI was checked from reliable sources. Six examples of compounds that are either removed from the CRC set or whose RI is corrected are given in Table 3. The first two compounds are discarded from the initial set, while the RI values of the following four are replaced by revised values.

From the first two rows of Table 3, it can be seen that dimethyl fumarate melts at 101.7 °C and 1,1-difluoroethane boils at −24 °C, and that their RI are measured at 110 °C and −72 °C, respectively. These two compounds cannot be kept in the final CRC set since their experimental RI is not taken at 20 °C. For (dichlorofluoromethyl)benzene diplayed in the third row, the given CRC value, measured at 11 °C, is replaced by a value that is an average of three measurements taken at 20 °C. An RI correction is also made for 1,1,1-trichloro-2,2,2-trifluoroethane (fourth row), whose CRC RI is measured at 35 °C. The small correction observed between the values measured at 20 and 35 °C (0.001) indicates that the CRC value is probably incorrect. The last two rows show two examples, glycerol 1-acetate and cyclohexylideneacetonitrile, for which large deviations in the predictions are observed with the GM24 model (−0.035 and −0.051, respectively). Consequently, their RI was checked in the literature from at least two different sources. It turns out that the values found either in Reaxys [32], CAS [35], or Landolt [34] are concordant but in disagreement with the values listed in the CRC database. The new values are then retained for the final RMSE test calculations. Out of the entire CRC set, approximately 20 compounds had their RI values corrected thanks to the discrepancies observed following the computations made with the GM24 model. All the replaced CRC RI values are used as is, i.e., without temperature correction. The maximum error for a compound whose RI is measured at 30 °C instead of 20 °C is about 0.005, which is of the same order as the GM24 model mean square deviation (Table 1, top row) and acceptable. From the initial CRC set, 259 compounds were discarded (leaving 1366 compounds), and 98 had their RI verified in the literature, resulting in the modification of 86 RI values. When only one RI value can be found in the literature and it disagrees with the CRC value, or when several values are found but differ from each other by more than 0.002, the initial CRC value is kept. Full details of the data reduction and RI value modifications for datasets are provided in the Appendix A section (spreadsheet tabs CRC-1366 and CRC-259 discarded).

#### 2.3.2. Performance Comparison of the Three Models

Table 4 gathers for the three datasets and the four models the root mean square errors computed with Equation (9), where $N_{T}$ represents the number of molecules under test. For the QSPR and GC models applied to their own set (HR-JT and CCAI, respectively), these quantities are called RMSTE since the compounds are members of the training sets, while in all other cases they are called test RMSE, the compounds being fresh data. Test RMSE is computed using Equation (9), in which ${RI}_{est.}^{i}$ is replaced by ${RI}_{pred.}^{i}$, which is the RI value predicted by the model for the molecule *i* of the test set.

In all cases, the graph machine-based model, with a test RMSE close to or equal to 0.010 (Table 4, last column), gives better estimations than the QSPR, GC, and GF models, for which the same test RMSE varies from 0.016 to 0.065. The GM24 model performs even better in testing than the QSPR and GC models do on training, since the values displayed in those cases are errors on the training set (Table 4, second row, columns 2 and 5, and third row, columns 3 and 5). It can also be seen that the Redmond model gives poor results on the Cai’s set, which contains molecules with halogen and nitrogen atoms, five types of atoms for which it has not been parameterized (Table 4, second row, second column). If the Redmond test set is restricted to 43 (out of 108) molecules that contain only carbon, hydrogen, and oxygen atoms, then the computed test RMSE drops to 0.026. This value is then comparable to the one computed for the 111 molecules in the original Redmond training set (0.022). Similarly, computing the test RMSE with the GC method for the remaining 40 molecules (out of 52) in the Redmond set yields an average value of 0.065 (Table 4, first row, third column), which is much higher than the RMSTE value computed for the 191 molecules in the original Cai training set (0.021). This is also why the RMSE for the CRC test cannot be processed with the Redmond and Cai models, as many atomic or group contributions needed to compute the molar polarizability of many compounds are not available. It is also noticeable that the most efficient of the three models that are compared to the GM model on the Redmond and Cai datasets is the GF model, with computed test RMSE values equal to 0.016 and 0.017 (Table 4, first and second rows, penultimate column).

Lastly, RI prediction results with graph machines and the GF model for the 1366 molecules in the CRC set are compared in Figure 4, which is the scatter plot of the results (predicted values versus measured values). Figure 4 shows that the graph machines lead to a more accurate prediction than the GF model, as the blue points further from the bisector of the graph than the red ones. It should be noted, however, that both models poorly predict the refractive index of trithiocarbonic acid (deviation 0.129 with GM24). Its refractive index has been measured by G. Gattow [36] and is found in several monographs since then [37,38]. No other refractive index value could be found in the literature for this acid. We did, however, retrieve many experimental refractive indices of various trithiocarbonates [39], which prompted us to test our model with a couple of these compounds. For example, prediction of the refractive indices of dimethyl trithiocarbonate and ethyl phenyl trithiocarbonate with the GM24 model leads to values equal to 1.631 and 1.636, respectively, which are not so far off the experimental values of 1.676 and 1.679, respectively. One possible explanation for the significant prediction error still observed for these two molecules (deviations of 0.045 and 0.043) is that no carbon atoms linked to three sulfur atoms are present in any of the molecules in the training set. In any case, it is more than likely that the value reported for trithiocarbonic acid is uncertain and should not be retained for a future model.

The details of the resulting RI predictions for those molecules are also available for download in the Appendix A (spreadsheet tab CRC-1366).

## 3. Materials and Methods

The first required step for machine learning simulation is the construction of the training set. To that end, several refractive index sources were used [20,32,34,35,40,41], the most important being the Tokyo Chemical Industry (TCI) dataset [20]. Thus, the training and test sets accessible in Bouteloup’s paper, made up of 3622 and 3621 liquids, respectively, were first used in the present study. These compounds were extracted from chemical supplier TCI’s 2018 online catalog because they had a measured refractive index (RI) there and because they were in the liquid state at 20 °C. The wavelength at which the measurements were made was not specified, but comparison of numerous values with those from reliable sources [32,34] confirms that it is that of the sodium D line, i.e., 589 nm.

Interestingly, the size of these datasets is much larger than the size (typically a few hundred compounds) of the datasets used in previous graph machine approaches [29,30,31]; in addition, the number of atom types (up to 16) is larger than the number of atom types present in previous graph machine approaches (up to 6).

As the range of refractive index values is quite narrow (1.296 to 1.687), it is important to find the appropriate balance between the accuracy of the measured values for a compound and their standard deviation when several experimental values are published. Analysis of several dozen compounds for which at least five different refractive index values were measured [34] reveals a standard deviation of measurement in most cases above 0.001. Therefore, all index values from now on will be rounded off to three decimal places. Finally, since the refractive index depends on both the experimental wavelength and the temperature, special care must be taken with these particular checks, namely 589 nm and 20 °C. However, as more and more refractive indices found in the literature are reported at 25 °C, a tolerance was granted in our data collection, leading to the use of uncorrected measurement values up to 30 °C. This rather rare practice, which only occurred if no data was available at the reference temperature, can introduce an error close to 0.1% of the mean measurement value, which is acceptable.

First of all, the lists of Bouteloup training and test sets, for which compounds were referenced solely by their CAS registry number (CASRN, abbreviated in RN), were supplemented with the isomeric SMILES code, molecular formula, and chemical name of all compounds. A simple script with the RN as input was used to download those identifiers from the PubChem database [42]. To add stereochemical information, if any, all the retrieved SMILES codes were then searched in the CAS database [35]. As a result, 588 compounds out of 7243 were retrieved with a stereochemical label in their isomeric SMILES as well as in their CAS names.

### 3.1. Data Analysis and Curation

The GM approach requires that the compounds whose RI is to be estimated have a well-defined structure (see Section 3.3). All data were carefully analyzed to eliminate any compound that could not be described by a unique chemical structure. In particular, 31 compounds that were retrieved from the CAS database as undefined substances, mainly polymers and mixtures, were replaced by compounds with similar structures, not already present in the datasets, but with known RI values. For example, the compound listed with RN 25513-64-8 is a 1:1 mixture of two diamines that have nearly identical RI values [32]. One of them, 2,2,4-trimethylhexane-1,6-diamine, was selected instead of the initial mixture. Besides those mixtures and polymers, 4-methoxycinnamonitrile was found to be solid at 20 °C, which prevents measuring its RI at this temperature. Therefore, it has been replaced by the liquid cinnamonitrile that has a similar structure and an RI available in the literature [34]. All the choices made for the 32 undefined substance replacements are detailed in the Appendix A.

While scanning the data for other compounds with ill-defined structures, the presence of several enantiomeric pairs was detected, often accompanied by the corresponding racemic mixture, also called racemate. This is a particular case since the two enantiomers and their racemate have theoretically the same RI. Consider the cases, for instance, of (*S*)-2-ethyloxirane and (*R*)-2-ethyloxirane, whose structures are shown in Figure 5. They are members of the initial training and test sets, respectively, and since they have structures that are nonsuperimposable mirror images of each other, they are enantiomers. The racemate 2-ethyloxirane, which belongs to the training set, is also represented in Figure 5: it is a 1:1 mixture of the two above enantiomers. As a racemate, its structure lacks a wedge bond, which indicates a precise configuration for the stereogenic center (noted with an asterisk), which is not the case for the two enantiomeric structures.

Now, as it is well known [43] that two enantiomers have identical physical and chemical properties, the (*S*)- and (*R*)-2-ethyloxirane should have the same RI. This implies that the stereochemical labels of their isomeric SMILES (@H and @@H in Figure 5) are not relevant for the graph machine construction. Because these stereochemical labels are the only difference between the isomeric SMILES of the enantiomers, ignoring them leads to using the same SMILES for both enantiomers, as well as for the racemate 2-ethyloxirane, as shown in Figure 5. Therefore, only one of the three compounds should be retained for training or testing, the other two being considered duplicates. In the following, a compound with a stereochemical label will be preferred over the racemate, even if only one enantiomer is present with the racemate. In the case exemplified above, the (*R*)-2-ethyloxirane with RN 3760-95-0 was retained with its isomeric SMILES in the final test set. The algorithm used for GM construction was thus designed to automatically recognize a compound with one stereogenic center as a potential enantiomer, hence discarding its stereochemical label before building its graph machine. The enantiomeric pairs were equally distributed between the initial training and test sets, and the selection of one enantiomer was performed based on the requirement that the final sizes of the training and test sets should be similar.

A further simplification results from the occurrence of diastereomeric compounds in the data, either with a *cis* or *trans* configuration as in cyclic compounds or with an *E* or *Z* configuration as in alkenes. Indeed, in this second particular case, compounds that are mixtures of the above pure *cis* and *trans*, or *E* and *Z*, compounds may be present in the sets. As these compounds are mixtures, their exact composition is unknown, so they cannot be kept in the final datasets. Thus, *trans*-1,2-dimethylcyclohexane, a member of the initial training set, and *cis*-1,2-dimethylcyclohexane, which belongs to the test set, are two diastereomers of the first category that are retained in the final sets. On the contrary, 1,2-dimethylcyclohexane, which is a *cis* and *trans* mixture, is eliminated. When compounds are present only as *E*/Z or *cis*/*trans* mixtures, such as 2-nonene or 1-bromo-2-fluorocyclohexane, they are kept as such, and their GM contains no stereochemical label. Finally, it should be noted that when the stereochemistry of these diastereomers is not considered, their SMILES are the same, so their estimated RI is the same while their measured RI is slightly different (up to 0.011).

Therefore, after the elimination of 105 enantiomers, 73 racemates, and 33 mixtures, the training and test sets used for model selection and evaluation contain 3516 and 3515 compounds, respectively. The two sets are available in the Appendix A, as well as the list of the withdrawn compounds, with a short explanation.

### 3.2. Analysis of Homologous Series

One way of checking the validity of some experimental data collected was, for homologous series, to plot the variation of the measured refractive index as a function of the number of carbon atoms in the compounds. The quality and regularity of the curves obtained prompted us to analyze this variation on the basis of Equation (4). Under the reasonable assumption that α and V are additive with respect to functional groups, for the particular case of homologous series (where the repeating unit is typically CH_2_, but could be any other repeating unit such as CF_2_ or Si(CH_3_)_2_O), it can be shown based on Equation (4) (cf. Appendix A, Section B) that the refractive index *n* follows the law given in Equation (10):(10)$$n=\sqrt{\frac{n_{repeat}^{2}N+B}{N+C}},$$
where *N* is the number of repeating units, $n_{repeat}^{2}$ is the squared refractive index for the infinite polymer composed from the repeating unit, and B/C is *n*^2^ for the molecule with no repeating unit (see last two columns of Table 5). We made two different comparisons. The first one only compares molecules with different functional groups, sharing CH_2_ as the repeating unit. In this case, $n_{repeat}^{2}$ is expected to be *n*^2^ for polyethylene. We thus used the known value $n_{repeat}$ = 1.476 [44]. For the five series chosen as an example, Equation (10) is closely followed, only fitting B and C (listed in Table 5) to experimental data (cf. Figure 6a, data points listed in Appendix A, Section C). Note that Equation (10) assumes that if the refractive index of the polymer composed of repeating units is equal to that of the molecule without repeating units, then the refractive index of the whole homologous series will be constant. This is approximately the case for the α,ω-diaminoalkanes in Figure 6a, since *n_PE_* is nearly equal to *n_hydrazine_*.

The second comparison introduces homologous series with different repeating units: *n*-alkanes, perfluoroalkanes, and methylated siloxanes (cf. Table 5 for the fitted parameters). Here, $n_{repeat}$ was fitted to experimental data, including *n*-alkanes for consistency. As shown in Figure 6b, the fitted values of $n_{repeat}$ that can be extrapolated from the data do differ substantially. This can be attributed to the different dispersive properties of CH_2_, CF_2_, and Si(CH_3_)_2_O functional groups.

Overall, this analysis illustrates that refractive index measurements on homologous series are consistent with fundamental physicochemical relationships and supports the hypothesis that refractive index data are an accurate experimental index for dispersive interactions.

### 3.3. Graph Machine Modeling

In graph machine-based models, molecules are described as graphs derived from their 2D structure, and the parameterized functions that compute the estimation of the property of interest, herein the refractive index, reflect the compound molecular structures. In the present case, a graph machine provides an estimate of the refractive index, which is a continuous quantity, indicating that the task is a regression task. The design of a graph-machine-based model includes the following steps:Construction of the 2D-graph of the molecule from its SMILES representation: each node of the graph is a non-H atom, and each edge of the graph is a chemical bond. Each node has at least two labels: the nature of the atom and its degree (the number of chemical bonds that bind it to its adjacent non-H atoms). For molecules that contain stereochemical information such as *E*/*Z* configurations, or wedge bonds, and hence R/S configurations, additional labels that we have named iso and chi are added to the relevant nodes. For molecules that contain cycles and are hence represented by a cyclic graph, one edge is deleted for each cycle of the molecule in order to form an acyclic graph in which every path of the graph ends at a specific node called the root or output node.Construction of the computational structure: for each acyclic graph, a function is generated by implementing, at each node of the graph, a parameterized nonlinear function called the node function, typically a multi-layer perceptron (MLP) with tanh activation functions for the hidden neurons and a linear output neuron. Since this construction does not require any descriptor, biases (neurons with non-trainable outputs equal to 1) are used instead of traditional inputs for the MLP, typically one for each label (e.g., Cl-h0, D3-h0, and iso1-h0 in Figure 7). The trick of this construction is to use the same function for all nodes and for all graphs. Therefore, the number of parameters in the resulting model is equal to the number of parameters in the chosen node function. As a result of this construction, the value computed by the output node of each model, which is intended to be an estimate of the refractive index, depends solely on the 2D structure of the molecule and the node function parameter values.Estimation of the parameters of the node function by training from the database: this is done by minimizing the sum of squared errors *J*(***θ***) defined in the next subsection.Additional details of the above steps are given in previous papers [28,45]. Examples of graph machine constructions that illustrate the previous conventions are shown in Figure 7 for (a) (*Z*)-1,2-dichloroethene and (b) (2*S*,3*S*)-2,3-dimethyloxirane, two compounds that contain stereochemical features.

Thus, in case (a) of Figure 7, the transformation of (*Z*)-1,2-dichloroethene into a directed graph in step ①, results in the two Cl atoms being mapped with the blue-green nodes of the atomic type label Cl and a degree label equal to 1. The *Z* configuration of the double bond is encoded by the stereochemical labels, respectively equal to iso1 and iso2. An *E* configuration for the same alkene would correspond to the pair {iso1, iso1}. As for the two nodes representing the olefinic carbon atoms bonded to one hydrogen, they have a C atom label and a degree equal to 3. The root node is assigned to one of them, and as such, it corresponds to the MLP that outputs the RI value in the GM obtained in step ②, shown at right in Figure 7a.

Case (b) in Figure 7 for (2*S*,3*S*)-2,3-dimethyloxirane is more complicated because the oxirane ring must be broken in step ① to obtain a directed acyclic graph. This results in a degree equal to 2 for the carbon-type nodes, which represent the two carbon atoms between which the bond is broken. Since those carbons were initially bonded to one hydrogen, their node degrees must equal 3 in the resulting graph. Increasing the node degrees to 3, while they are only connected to two other nodes, indicates that a disconnection has occurred. Finally, to account for the *trans* configuration of the two methyl substituents, the stereochemical label chi1 is added to these same two nodes. The latter methyl groups are transformed into nodes that also have a carbon-like label but a degree label equal to 1. To complete the directed acyclic graph description, the root node maps to the oxygen atom and thus has a degree of 2 and an O atom type. Consequently, the GM shown at right in Figure 7b, obtained after step ②, has an output computed by the MLP implemented on the O-type root node. The (2*S*,3*S*)-2,3-dimethyloxirane diastereomer, i.e., the meso (*R*,*S*)-2,3-dimethyloxirane, would require the pair {chi1, chi2} to encode the *cis* configuration of the oxirane substituents.

Finally, note that in both GMs, the MLP sequence, symbolized by the black, blue-green, and yellow disks, exactly mirrors the two starting structures, with the exception of the bond that was broken in dimethyloxirane to obtain the directed acyclic graph.

### 3.4. Model Selection

For graph machines as well as for any other machine-learning model, this step is particularly important. Its purpose is to determine, given the data available, the model complexity that will result in the best generalization. Indeed, a model that is not complex enough is unable to fit the data, and therefore to generalize, while a too complex model overfits the data and predicts poorly. For graph machines, the number of adjustable parameters depends on the number of hidden neurons present in the MLP that has been used to design them, so finding the optimal complexity will consist in finding the number of hidden neurons that ensures the best model generalization capabilities. This task is performed by first partitioning the available data into a training/validation set for designing and selecting the model, a training set for simplicity, and a test set for estimating the generalization error of the selected model.

In what follows, the set of graph machines that are built from the learning examples will be referred to as the graph machine-based model. The parameters of these models are estimated, given a training set of *N_T_* elements, by minimizing, using the weight sharing method between all nodes of all graph machines, the sum of squared errors of the cost function *J*(***θ***) (Equation (11)):(11)$$J(\theta )=\sum_{i=1}^{N_{T}}{\left(y_{i}-g_{i}(\theta )\right)}^{2}$$
where *y_i_* is the measured value of the RI for the *i*-th element of the training set, ***θ*** is the vector of parameters, and *g_i_*(***θ***) is the value of the RI estimated by the graph machine for that element. In this work, *g_i_*(***θ***) is constructed as a combination of MLPs with a single hidden layer that reflects the graph structure of the *i*-th element. This MLP is a linear combination of nonlinear functions called hidden neurons, which are the hyperbolic tangent functions of a linear combination of the variables. All minimizations of the cost function are performed by the Levenberg–Marquardt algorithm, which is well suited to optimization problems with a moderate number of variables [46].

The estimation of generalization error for model selection is then performed by computation of the virtual leave-one-out (VLOO) score, which is known to be a first-order approximation of the leave-one-out (LOO) score but which is obtained at a much smaller computational cost with a dataset of 3516 examples [47]. This is due to the fact that the computation of the VLOO score involves the training of a single graph machine-based model containing *N_T_* graph machines, while the LOO score computation requires the training of 3516 graph machine-based models containing each *N_T_*−1 graph machine. As explained in a previous paper [30], this score relies on a first-order approximation of the estimation error that would be obtained on each molecule of the training set if that molecule had been removed from that set before training. Thus, denoting by ***θ****_m_* the parameter vector after completion of training, the VLOO score (Equation (12)) is defined as the root-mean-square of the VLOO prediction errors:(12)$$VLOO\;score=\sqrt{\frac{1}{N_{T}}\sum_{i=1}^{N_{T}}{\left(y_{i}-{g_{i}(\theta }_{m}^{–i})\right)}^{2}}\;,$$
where ${g_{i}(\theta }_{m}^{-i})\;$is a first-order approximation of the predicted RI value of element *i* provided by the *i*-th graph machine when the latter is not present in the training set, and *y_i_* is the measured value of the RI for the *i*-th element of the training set. The VLOO score is consequently an estimate of the model’s generalization error. A detailed mathematical analysis of VLOO is provided by Monari and Dreyfus [47].

LOO scores and VLOO scores are the tools used to select the complexity of the model. When the complexity (i.e., number of hidden neurons) increases, the root mean square training error (RMSTE) decreases, and VLOO reaches a floor value. If overtraining occurs, then VLOO increases.

Our training procedure is, then:Launch a large number (e.g., 100) of parameter initializations followed by a full training computation;Select a small number (e.g., 10) of results with the smallest VLOO values;Use these selected models to compute the average property estimation for a fresh example.

This procedure avoids the occurrence of overtraining.

With the present training set of 3516 molecules, the VLOO score is computed instead of the LOO score, due to the excessive computational load that would be required in the latter case. For each complexity, 100 trainings are performed with different initial parameter values, so that the mean of the ten smallest VLOO scores is computed for the selection of the most appropriate complexity.

Once the complexity of the graph machine-based models is selected, it is applied to the test set of 3515 molecules. The graph machine of each test set molecule is then constructed as explained above, and the parameters of the model (***θ****_m_*) are assigned to its node functions so that the graph machine output provides an estimate of the RI for that molecule. The true benefit of this approach is the absence of descriptors; the SMILES codes are the only required information. Moreover, the same set of graph machines can be reused to estimate another property or activity after a re-training of the model. More details on graph machine construction and training can be found in an earlier paper [28].

## 4. Discussion

So far, the selected GM24 graph machine model (Section 2.1) has performed very well compared to all other published models. It has also been more than twice as efficient as the GF model for the RI prediction of the 3515 compounds in the TCI test set. Thus, a test RMSE equal to 0.006 (Table 2, bottom row, column 2) was obtained for the graph machine-based model on this set, while a value equal to 0.014 was computed for the GF model. There are several possible explanations for this difference. First of all, while the GF model uses only 89 parameters (see Introduction), the GM24 model counts 761 parameters, which allows it to account more effectively for the non-linearity of the RI property to be predicted. Contrary to what was announced by the designers of the GF model [20], the graph machine-based model is not prone to overtraining, as explained in Section 3.4 and as shown by the results in Table 4 (last column) for all test datasets. In fact, a graph machine model with 121 parameters for a training set counting 300 examples has already been used before, with no evidence of overfitting the data [30]. However, the ratio of the training set size to the number of model parameters was equal to 2.5, whereas it is 4.6 for the present dataset, which should rule out such an overtraining behavior. A second reason would be that the Lorentz–Lorenz equation, which is at the heart of the GF method and links the refractive index, the molar polarizability, and the molar volume for a given liquid, is not verified beyond a certain accuracy [48]. Thirdly, the estimation of refractive index with the GF model relies on experimental values from two series of measurements for the training set compounds, which increases the risk of errors arising from the bibliographic sources used and the experimental measurements. Lastly, the graph machine approach is based on the construction of parameterized functions from the topological information contained in the supplied SMILES codes. As a result, these functions reflect the molecular structures of the compounds that were used to build them. This is very important because it allows them to do more than just count and add atomic contributions; for example, they can encode the configuration of a carbon atom when it is bonded to four different atoms or connected by a double bond to another atom with two possible configurations. On the contrary, the GF model cannot discriminate between diastereomers, giving identical RI predictions for such isomers. For example, an RI value of 1.466 is predicted by the GF model for the 1,3-dichloroprop-1-ene *E* and *Z* isomers, while their measured RI values are respectively equal to 1.475 and 1.470. In this particular case, the GM24 model computed, as expected, two different values equal to 1.472 and 1.468, more in line with the experimental measurements.

However, while still good, the results are more mixed for the CRC, Redmond, and Cai datasets than for the TCI set. For example, in the case of the former set, 11 molecules are predicted with a deviation of more than 0.040 in absolute value, compared to only 5 molecules in the case of the TCI test set, which has 2.5 times more compounds. Moreover, the RI predictions of the diastereomeric pairs are not better than those for compounds without stereochemistry; the test RMSE computed for the corresponding 128 and 1201 compounds are both equal to 0.010. This is not completely surprising since the training set used to build the GM24 model has only 157 molecules with stereochemical information, of which only 18 are grouped into diastereomeric pairs, mostly compounds containing *cis*-*trans* isomerism.

To address these shortcomings, a larger training set with more diastereomeric pairs was built. All the previous sets were added together to produce a file containing no less than 8397 compounds. Once the duplicates had been removed, it contained 8267 compounds, including 473 diastereomers, 185 of which have at least one diastereomer present in the training set. In addition, to assess the predictive capability of this new model, a test set of 175 compounds containing 15 atom types and a large percentage of diastereomers was designed from a variety of reliable sources [34,37,38,40,41], hence the name MIX test. Only those compounds for which RI values from at least two different sources were concordant were retained. Details on how the training and test datasets were designed and which RI values were modified in them are provided in the Appendix A section (spreadsheet tabs named Training-8267 and Removed Compounds-118).

The set of 8267 compounds was trained under the usual conditions with a graph machine-based model having 24 neurons in the hidden layer (see Section 2.1) to produce the results reported in Table 6.

A quick comparison with those reported in Table 2 shows that increasing the number of training examples from 3616 to 8267 only slightly alters the training and prediction results. In fact, the deviations obtained for the MIX test compounds are even smaller (e.g., −0.022 vs. −0.051), despite the fact that some of them, such as diethylsilane or 5,5-dimethyl-3,4,5,6-tetrahydro-1H-germolo[3,4-c]thiophene, have a structure that differs slightly from that of the training set compounds. The test RMSE value of 0.007 is still very good considering the wide variety of different atoms (15 out of 16) in the molecules tested. In addition, RI prediction for diastereomers is also very good, even better than for the TCI test isomers, with the test RMSE also being lower than the previously obtained value (0.005 vs. 0.006), with about the same number of diastereomers (38 vs. 43).

Figure 8 shows a scatter plot of the prediction results with the GM24 model for the 175 compounds of the MIX test. The fit is very good, especially for diastereomeric compounds, which are represented by red disks. This figure clearly shows that the RI predictions for each compound in a diastereomeric pair, or array if several stereogenic centers are present, are slightly different. In addition, the ranking of the values of these pairs is usually well respected in the case of predicted values. For example, for *cis*- and *trans*-2,4-pentadienenitrile, the experimental values are 1.486 and 1.499, respectively, and the predicted values are 1.491 and 1.493 in the same order. Similarly, (1*S*,3*S*)- and (1*R*,2*s*,3*S*)-1,2,3-trimethylcyclopentane have experimental RI values of 1.422 and 1.425, while their predicted RI values are 1.425 and 1.428, also in the same order. Hence, we can already conclude that this first approach to processing the prediction of refractive indices of diastereomeric compounds is working successfully.

On the other hand, the estimations performed with the GF model for the MIX test compounds (not represented here) are less accurate, as the RMSE test value is equal to 0.017. Furthermore, prediction is not possible for 5 of the 175 compounds, as the parameters required for calculation, namely *V*_Si42_, *R*_Si42_, *V*_P31_, *R*_P31_, *V*_P32_, and *R*_P32_, are not available. This situation does not arise with graph machines if the atom is already present during training, like Si or P in the cases above. If the atom is absent from the training set, a simple re-train after adding to the training set a few compounds with known refractive indices and containing the missing atom can then secure the GM24 prediction.

To further test the robustness of our model, we have built up a small database of 22 exotic liquids, for which we have also retained at least two matching values for each measured refractive index. Despite the presence of atoms absent from the training set (Al, As) and molecules containing no carbon atoms (water, phosphorus tribromide, trichlorosilane), all refractive indices were estimated, with the parameters of the missing atoms being assigned a zero value. However, an unusually high RMSE value (0.036) was computed for this test, but this was reduced to 0.020 when arsenic trichloride was removed from the set. In any case, this final experiment shows that the graph machine-based model used to estimate the refractive index is particularly robust, since it has not been possible to fault it. The results computed for the exotic set of compounds are also available for download in the Appendix A.

As a result, we have developed a demonstration tool based on Docker technology and fed with the built-in data (refractive indices at 20 °C and SMILES). It allows you to replicate the predictions for the 175 compounds on the test set. In addition, the demo software version 1.0 is also able to perform the prediction with good accuracy of the refractive index of any liquid of molecules containing carbon, hydrogen, oxygen, nitrogen, halogens, sulfur, boron, silicon, phosphorus, titanium, selenium, tin, and germanium atoms, starting from its SMILES code. Details on how to install Docker, download, and use our demo are available in the Appendix A. Readers are then welcome to use the demo software (v. 1.0) to estimate the refractive index of the liquids of interest to them.

## 5. Conclusions

The estimation of the refractive index of liquids from the structure of their constituent molecules attracted much attention due to the importance of that property in a variety of scientific and technical areas. The present article reports four main innovations: (i) the estimation of the refractive index of organic liquids by graph machines, a machine learning method that allows the estimation of properties or activities of molecules directly from their structure described by their SMILES codes, without requiring any other descriptors, (ii) the graph machine method is applied to a set of several thousand compounds and can distinguish diastereomers, predicting different indices for each of them; (iii) the comparison of the accuracy of refractive index predictions obtained by several methods (QSPR, group contribution, geometrical fragment, and graph machines); and (iv) software (v. 1.0) is available for download to predict the refractive index of a liquid compound from its SMILES code.

A database of 3516 organic compounds is used for training and model selection, and a database of 3515 compounds is used for testing. Graph machines, which perform regression from the graphs derived from the SMILES codes, are first constructed. The graph-machine-based models are trained, and a model selection is performed by virtual Leave-One-Out (VLOO) to select a node function complexity of 24 neurons. The resulting root-mean-square error on the test set using this complexity is then equal to 0.006.

For comparison, when applied to fresh datasets containing 108 and 40 compounds, respectively, the QSPR method underestimates the refractive index of the test set by a large amount, with a root-mean-square estimation (RMSE) error of 0.062, while the group contribution method overestimates it, with an RMSE of 0.065. Conversely, the geometrical fragment method underestimates only slightly the refractive index of both sets, with a smaller RMSE of 0.016, which is also the RMSE value computed with the larger CRC test set. For all sets, the graph machine-based model gives a pretty constant RMSE value around 0.010 and does not underestimate or overestimate the refractive index of the datasets.

A final graph machine-based model, with the same complexity as the previous model, is trained on a large set of 8267 compounds for which refractive indices measured at 20 °C and 589 nm have been compiled. Successfully tested on a set of 175 different compounds for which refractive indices were also carefully verified, and then on a set of 22 exotic compounds containing mostly no carbon atoms, this model has proved particularly robust and reliable. In particular, the accuracy achieved enables us to differentiate and classify the refractive indices of diastereomeric compounds. Its main limitation lies in the structures used as inputs for the construction of the graph machines themselves: if a structural form such as an enol is favored, then using a ketone form will lead to an underestimation of the refractive index. In such a case, it is useful to calculate the percentages of the two tautomeric forms with software like COSMO-RS (release 2023) in order to use the SMILES code of the more stable form.

The current results provide new evidence of the ability of graph machines to accurately estimate the properties of molecules from their structures, especially when the property to be estimated, such as the refractive index, is topology-based, provided that the information contained in the S code is sufficiently relevant. In a forthcoming article, we will show that it can be extended to the prediction of an atomic property, as graph machines are also capable of predicting the chemical displacement of a carbon atom, a property that is highly dependent on the neighborhood of the atom under consideration.

For easy duplication of the presented results and for testing of the method on other molecules belonging to the same families as those present in our database, demonstration software (v. 1.0) is made available in the Appendix A, Sections E and F, and two videos explaining how to use it are provided on YouTube at https://youtu.be/BhAyUBkv7cM for Windows (accessed on 21 September 2023) and https://youtu.be/fe8kkQgsGOc for Macintosh (accessed on 21 September 2023).

## Figures and Tables

**Figure 1 molecules-28-06805-f001:**
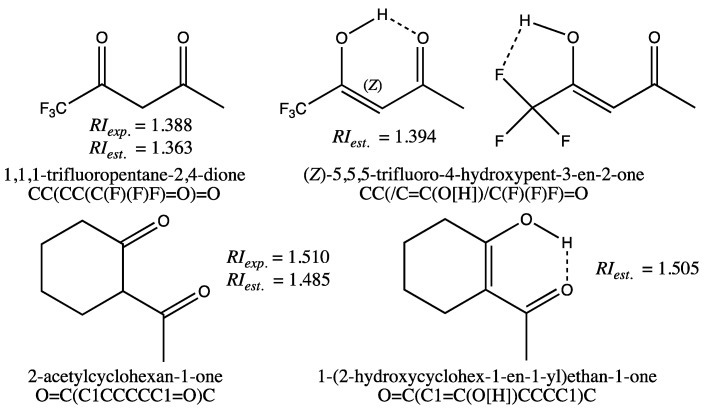
Structure of 1,3-diketones and keto-enol forms for 1,1,1-trifluoropentane-2,4-dione and 2-acetylcyclohexan-1-one, and SMILES codes used for RI computations with the GM24 model.

**Figure 2 molecules-28-06805-f002:**
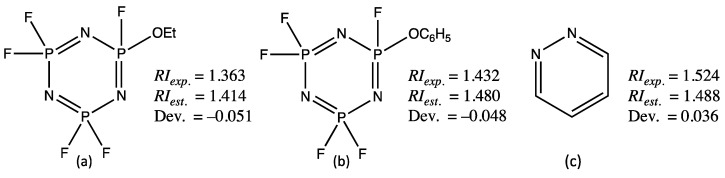
Structures (**a**–**c**) of the test set compounds that have the largest negative and positive deviations for their computed RI using the GM24 model. *RI_exp_*., *RI_est_*. and Dev. stand for experimental RI, estimated RI, and deviation.

**Figure 3 molecules-28-06805-f003:**
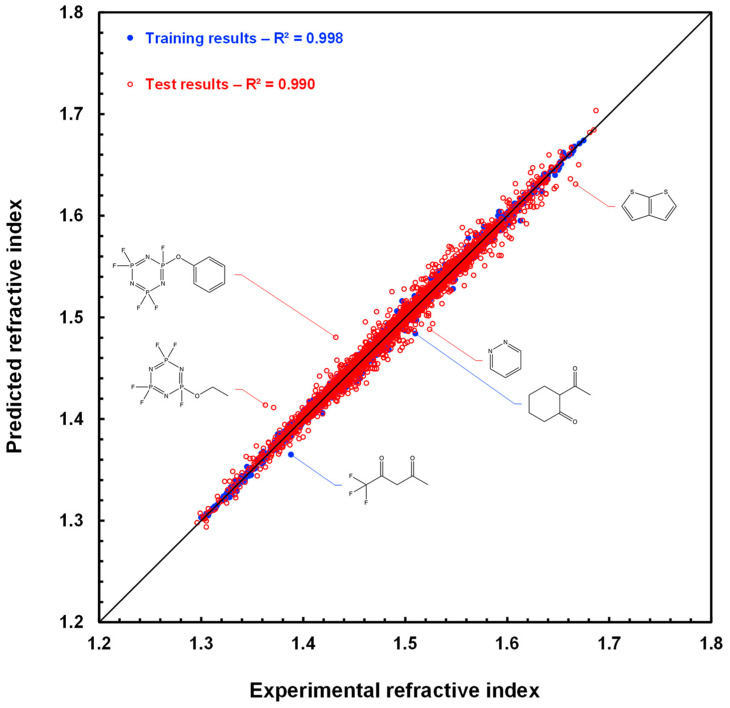
Scatter plot of refractive index estimations for the 3516 molecules of the TCI training set (blue disks) and of refractive index predictions for the 3515 molecules of the TCI test set (red circles) computed by graph machines vs. measured refractive index values. The black line is the bisector of the plot.

**Figure 4 molecules-28-06805-f004:**
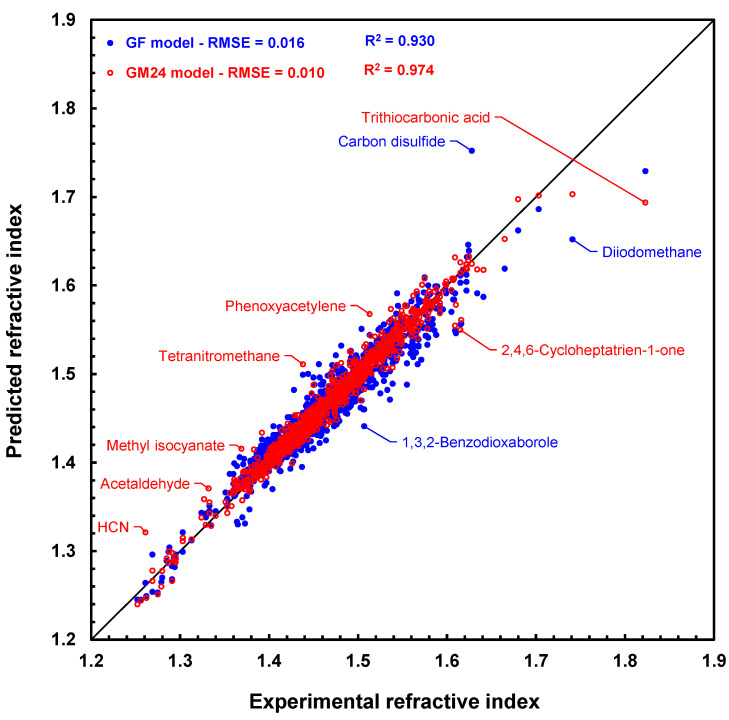
Scatter plot of refractive index predictions computed by geometrical fragment method [20] (blue disks) and graph machines (red circles) vs. measured refractive index values for the 1366 molecules in the CRC test set. The black line is the bisector of the plot.

**Figure 5 molecules-28-06805-f005:**
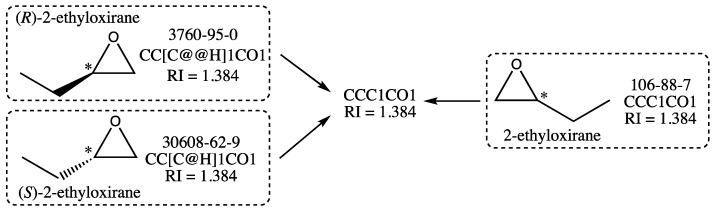
Example of dataset simplification for the three 2-ethyloxiranes shown with their structure, registry number, isomeric SMILES, and refractive index value. The stereogenic center is marked with an asterisk.

**Figure 6 molecules-28-06805-f006:**
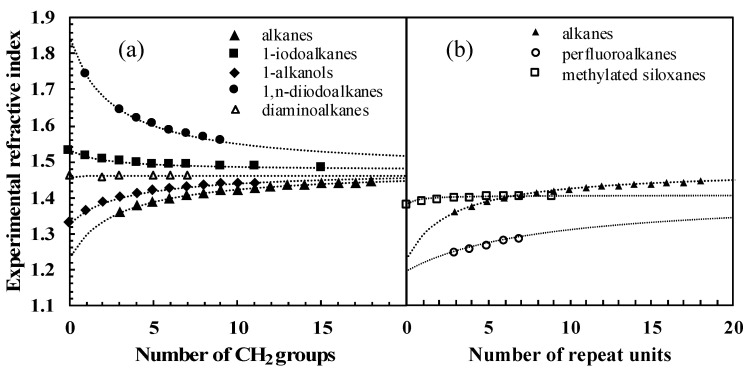
Refractive index vs. number of (**a**) CH_2_ repeated groups for five homologous series, (**b**) repeat units for three homologous series. Experimental data were extracted from [32,34,35,38]. The dotted lines were drawn using Equation (10).

**Figure 7 molecules-28-06805-f007:**
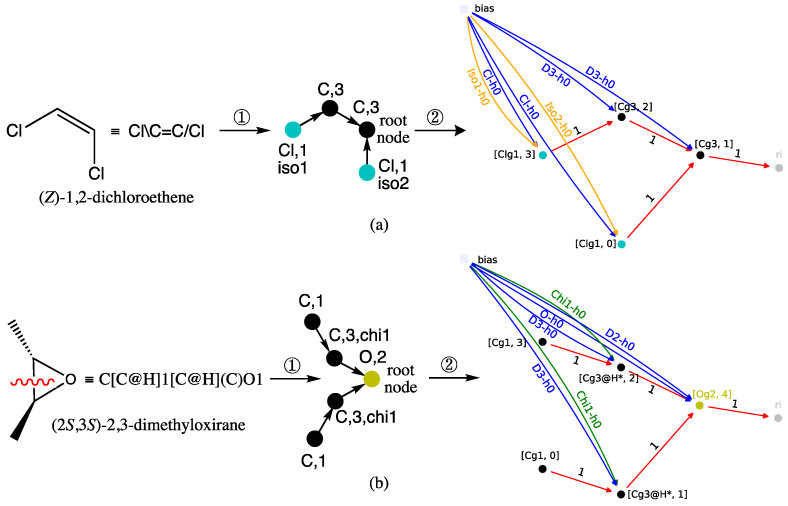
Coding of (**a**) (*Z*)-1,2-dichloroethene and (**b**) (2*S*,3*S*)-2,3-dimethyloxirane from their 2D-structure into their directed graph (①) and graph machine (②). To simplify the GM representations, some bias inputs are omitted, and the implemented node functions are MLPs with zero hidden neurons. The red wavy line indicates a cycle opening in step ① to obtain an acyclic graph. The asterisks on the nodes of graph machine (**b**) correspond to the carbon atoms between which a bond has been broken.

**Figure 8 molecules-28-06805-f008:**
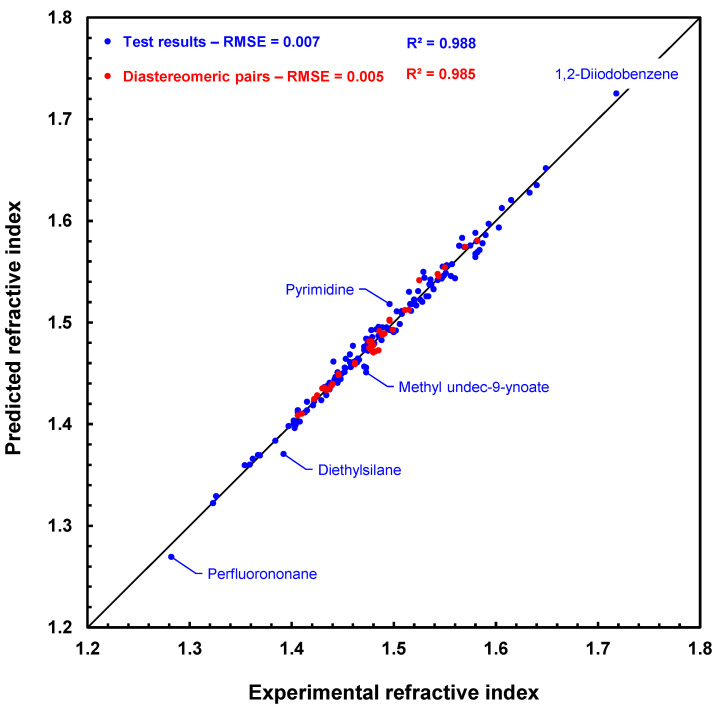
Scatter plot of refractive index predictions computed by graph machines vs. measured refractive index values for the 175 compounds of the MIX test. Red disks are for pairs of diastereomers, and blue disks are for other compounds. The dashed line (y = 0.998x + 0.004) is the regression line for the total set.

**Table 1 molecules-28-06805-t001:** Estimation of the refractive index from SMILES by graph machine-based models of increasing complexity.

Number of Hidden Neurons	6	8	10	12	14	16	18	20	22	24	26
RMSTE (10^−3^) ^1^	11	10	9	8	7	6	6	5	5	4	4
VLOOs (10^−3^) ^2^	12	11	10	9	8	8	7	7	7	6	6
MIN (10^−3^) ^3^	−47	−42	−41	−40	−39	−38	−33	−29	−24	−19	−18
MAX (10^−3^) ^3^	103	87	72	60	46	37	32	30	29	26	24

^1^ Mean of the RMSTE values and ^2^ mean of the VLOO scores, averaged over the 10 trained models (out of 100) having the smallest VLOO scores, both computed for three different parameter initializations, for the 3516 molecules of the training set; standard deviation for all means is smaller than 10^−4^, and ^3^ MIN and MAX denote the means of the maximum and minimum deviations from experiment.

**Table 2 molecules-28-06805-t002:** Performance of the GM24 model for TCI training and test sets.

Dataset	$N_{T}$ ^1^	RMSE ^2^	R^2^	MIN ^3^	MAX ^3^	STE ^4^
Training	3516	0.003	0.998	−0.019	0.026	0.002
Test	3515	0.006	0.990	−0.051	0.036	0.006

^1^ Number of elements in datasets, ^2^ root mean square error averaged over the 10 trained models (out of 100) having the smallest VLOO scores for the 3516 molecules of the training set, ^3^ minimum and maximum deviations from experiment and ^4^ root mean square error computed for compounds with a stereochemical label in their graph machine.

**Table 3 molecules-28-06805-t003:** Examples of compounds removed from the CRC test set or with RI corrected.

Compound Name	CRC *RI_exp_*. ^1^	GM24 *RI_pred_*. ^2^	Other Sources RI ^3^	Revised RI	MP or BP (°C) ^4^
Dimethyl fumarate	1.406 (110)	1.443	1.406@111 [34]	-	101.7 (mp)
1,1-Difluoroethane	1.301 (−72)	1.271	1.301@–72 [35]	-	−24 (bp)
(Dichlorofluoromethyl) benzene	1.518 (11)	1.514	1.514@20 [34] 1.513@20 [35]	1.514	liq.
1,1,1-Trichloro-2,2,2-trifluoroethane	1.361 (35)	1.365	1.360@20 [34] 1.360@20 [35]	1.360	liq.
Glycerol 1-acetate	1.416 (20)	1.451	1.450@20 [34] 1.450@20 [35]	1.450	liq.
Cyclohexylidene- acetonitrile	1.438 (25)	1.489	1.483@25 [32] 1.483@25 [35]	1.483	liq.

^1^ Values in brackets correspond to the measurement temperatures in °C, ^2^ predicted RI using graph machine-based model at 20 °C, ^3^ @T means measured at T °C as found in cited sources, and ^4^ Liq. indicates that the compound is a liquid at 20 °C.

**Table 4 molecules-28-06805-t004:** Test RMSE computed with the GM24 model and models designed by other authors.

Dataset	$N_{T}\;$ ^1^	Test RMSE (10^−3^)			
		QSPR	GC	GF	GM24 ^2^
HR-JT	52	*20* ^4^	65 ^3,5^	16	10
CCAI	116	62 ^3,6^	*14* ^4^	17	11
CRC	1366	-	-	16	10

^1^ Number of elements in datasets, ^2^ test root mean square error averaged over the 10 trained models (out of 100) having the smallest VLOO scores for the 3516 molecules of the training set, ^3^ test RMSE are computed only for molecules that are not present in the training sets used for model parameterization, that is, 40 and 108 molecules for the HR-JT and CCAI sets, respectively, ^4^ values in italics are RMSTE instead of test RMSE, ^5^ GC predictions are from the paper of Cai et al. [19], and ^6^ QSPR predictions are calculated with the equation given in the Redmond and Thompson paper [17].

**Table 5 molecules-28-06805-t005:** Fitted $n_{repeat}$, B and C coefficients from Equation (10) based on refractive indices for 7 homologous series.

Homologous Series	$n_{repeat}$	B	C	Initial Member (With No Repeat Unit)	$n_{exp.}$	$\sqrt{B/C}$
*n*-Alkanes	1.469 ^1^ ($n_{PE}$)	4.786	3.139	ethane	n/a (gas)	1.235
1-Iodoalkanes		4.837	2.063	iodomethane	1.531	1.531
Primary alcohols		6.008	3.407	methanol	1.329	1.328
Diaminoalkanes		207.670	97.681	hydrazine	1.457	1.458
Diodoalkanes		7.758	2.281	I_2_	n/a (solid)	1.844
*n*-Alkanes	1.475 ^2^	4.556	3.011	ethane	n/a (gas)	1.230
perfluoroalkanes	1.441	19.667	13.829	perfluoroethane	n/a (gas)	1.193
methylated siloxanes	1.398	2.021	1.067	hexamethyldisiloxane	1.377	1.376

^1^ $n_{repeat}$ coefficient (corresponding to the refractive index of polyethylene *n_PE_*) fitted for all five homologous series and ^2^ $n_{repeat}$ fitted from experimental data instead of assumed equal to that of polyethylene.

**Table 6 molecules-28-06805-t006:** Performance of the GM24 model for the final sets.

Dataset	$N_{T}$ ^1^	RMSE ^2^	R^2^	MIN ^3^	MAX ^3^	STE ^4^
Training	8267	0.004	0.995	−0.024	0.028	0.003
MIX Test	175	0.007	0.988	−0.022	0.022	0.005

^1^ Number of elements in datasets, ^2^ root mean square error averaged over the 10 trained models (out of 100) having the smallest VLOO scores for the 8267 molecules of the training set, ^3^ minimum and maximum deviations from experiment, and ^4^ root mean square error computed for the 22 compounds that have a stereochemical label in their graph machine.

## Data Availability

Not applicable.

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
