# Peer review of "Fast and Accurate Prediction of Refractive Index of Organic Liquids with Graph Machines"

_molecules, 2023, doi:10.3390/molecules28196805_

Round 1
Reviewer 1 Report
This article presents an interesting approach, the Graph Machines Model (GMM), which serves as a general (and advanced) model (or set of models) to predict the refractive index (RI) for various liquid organic compounds. Based on the results, the strength of the GMM lies in its ability to handle complex molecular structures containing different types of atoms, enabling accurate RI predictions for compounds of varying complexity (including stereoisomers). The model was based on a huge number of molecules and tested on a large data set. The authors demonstrate the superiority of the GMM by comparing its predictive capabilities with several fragment-based approaches found in the literature. A notable contribution of this work is the development of a Docker-based tool for predicting the RI of organic compounds with SMILES.
In addition, the manuscript is of excellent technical quality and very good clarity of presentation. Therefore, I believe that the article deserves to be published without changes.
Questions to the authors:
- Does the process of generating the graph machine algorithm depend on using canonical SMILES strings as input, or can any valid SMILES string be utilized, with the model subsequently converting it to canonical form?
- In the development of your model, did you explore the possibility of establishing a norm for the applicability domain of the predictive model? If not, have you identified specific types of molecules, compounds, or structural features that might be more susceptible to larger or smaller prediction errors? How can one assess this beforehand?
- The inclusion of a demonstration tool for property prediction is a valuable addition to the paper. Have you considered the feasibility of creating a web-based version of this tool? While the concept is straightforward, we understand that implementation may present challenges. Nonetheless, offering a web-based tool could significantly enhance accessibility and usability for a broader user base.
Reviewer 2 Report
The work is well structured, includes relevant references, and deal with a quite large and depurated database. The use of learn machines is well implemented and the results are sound. My only concern is related to the way the authors present the Virtual Leave-one-out (VLOO) procedure. Maybe the presentation of quite fast. I propose to enlarge the presentation of the concept and link in a more explicit way to equation (12). It seems that no data (now a single example) is provided comparing a VLOO result with a LOO one. This information could be useful for the reader.
